# Evaluation of Commercially Available Viral Transport Medium (VTM) for SARS-CoV-2 Inactivation and Use in Point-of-Care (POC) Testing

**DOI:** 10.3390/v12111208

**Published:** 2020-10-23

**Authors:** David van Bockel, C. Mee Ling Munier, Stuart Turville, Steven G. Badman, Gregory Walker, Alberto Ospina Stella, Anupriya Aggarwal, Malinna Yeang, Anna Condylios, Anthony D. Kelleher, Tanya L. Applegate, Andrew Vallely, David Whiley, William Rawlinson, Phillip Cunningham, John Kaldor, Rebecca Guy

**Affiliations:** 1Kirby Institute for Infection and Immunity in Society, UNSW Medicine, UNSW Sydney, Kensington, NSW 2052, Australia; cmunier@kirby.unsw.edu.au (C.M.L.M.); sturville@kirby.unsw.edu.au (S.T.); sbadman@kirby.unsw.edu.au (S.G.B.); aospinastella@kirby.unsw.edu.au (A.O.S.); aaggarwal@kirby.unsw.edu.au (A.A.); akelleher@kirby.unsw.edu.au (A.D.K.); Tapplegate@kirby.unsw.edu.au (T.L.A.); Avallely@kirby.unsw.edu.au (A.V.); Jkaldor@kirby.unsw.edu.au (J.K.); Rguy@kirby.unsw.edu.au (R.G.); 2NSW Health Pathology, Prince of Wales Hospital, Randwick, NSW 2052, Australia; gregory.walker@unsw.edu.au (G.W.); Malinna.Yeang@health.nsw.gov.au (M.Y.); Anna.Condylios@health.nsw.gov.au (A.C.); w.rawlinson@unsw.edu.au (W.R.); 3NSW State Reference Laboratory for HIV-AIDS/St Vincent’s Hospital Sydney, St Vincent’s Centre for Applied Medical Research, St Vincent’s Hospital Sydney Limited, Darlinghurst, NSW 2010, Australia; d.whiley@uq.edu.au; 4Australia Pathology Queensland, Royal Brisbane and Women’s Hospital, Herston, QLD 4006, Australia; 5Centre for Clinical Research, The University of Queensland, Royal Brisbane and Women’s Hospital Campus, Herston, QLD 4006, Australia; Philip.Cunningham@svha.org.au

**Keywords:** SARS-CoV-2, inactivation, virucidal, diagnostic

## Abstract

Critical to facilitating SARS-CoV-2 point-of-care (POC) testing is assurance that viruses present in specimens are inactivated onsite prior to processing. Here, we conducted experiments to determine the virucidal activity of commercially available Viral Transport Mediums (VTMs) to inactivate SARS-CoV-2. Independent testing methods for viral inactivation testing were applied, including a previously described World Health Organization (WHO) protocol, in addition to a buffer exchange method where the virus is physically separated from the VTM post exposure. The latter method enables sensitive detection of viral viability at higher viral titre when incubated with VTM. We demonstrate that VTM formulations, Primestore^®^ Molecular Transport Medium (MTM) and COPAN eNAT™ completely inactivate high-titre SARS-CoV-2 virus (>1 × 10^7^ copies/mL) and are compatible with POC processing. Furthermore, full viral inactivation was rapidly achieved in as little as 2 min of VTM exposure. We conclude that adding certain VTM formulations as a first step post specimen collection will render SARS-CoV-2 non-infectious for transport, or for further in-field POC molecular testing using rapid turnaround GeneXpert platforms or equivalent.

## 1. Introduction

The current method to detect the presence of SARS-CoV-2 virus during infection is reverse transcription polymerase chain reaction (RT-PCR), a molecular diagnostic testing technique, which detects the virus genetic material from respiratory samples. The World Health Organization (WHO) recommends that any testing for the presence of SARS-CoV-2 should be performed in appropriately equipped laboratories [1,2]. However, in March 2020, the Xpert^®^ Xpress SARS-CoV-2 assay [3] was approved by the regulatory bodies United States Food and Drug Administration (United States of America) and Therapeutic Goods Administration (Australia), enabling RT-PCR in the laboratory and point-of-care (POC) in decentralized settings [4]. The Xpert^®^ Xpress SARS-CoV-2 assay has similar accuracy to other RT-PCR assays but provides a result in 45 min or less using an early assay termination feature if a sample is positive, and is, therefore, able to facilitate timelier clinical and public health responses. Moreover, POC testing also provides an important opportunity to expand validated testing methods in low-and-middle-income countries (LMICs) where there is limited laboratory infrastructure [5], and also in remote areas of high-income countries where there are significant distances to city-based laboratories, such as Aboriginal communities in remote locations throughout Australia.

The WHO recommends that initial processing (before inactivation) of respiratory specimens should take place in a biological safety level (BSL2) cabinet (WHO laboratory safety manual 3rd edition; www.who.int/publications-detail/laboratory-biosafety-guidance-related-to-coronavirus-disease-(Covid-19)). However, more recently the guidance has extended to supporting POC testing platforms such as the GeneXpert, with similar advice available from the Centre for Disease Control and Prevention (CDC) and Public Health Laboratory Network (PHLN) in Australia (www.health.gov.au/resources/publications/phln-guidance-on-laboratory-testing-for-sars-cov-2-the-virus-that-causes-covid-19). The Xpert^®^ Xpress SARS-CoV-2 POC collection and testing process involves (i) collecting and placing a nasopharyngeal swab (NPS) in a viral transport medium; (ii) a minimal number of sample processing steps, such as agitating and standing the sample tube; (iii) a 300 μL aliquot is pipetted into a closed cartridge system; and (iv) the detection of amplified viral genetic material (RNA). The cartridge contains a chaotropic sample processing reagent that is mixed with the sample within five minutes of starting the test and commences denaturation. For these types of assays, the WHO considers sample manipulation, and the level of aerosol generation is minimal. The WHO guidelines recommend POC testing, or near-POC testing such as a risk-assessed benchtop workflow without the need for a BSL2 cabinet (WHO interim guidance, 13 March 2020). These protocols have been successfully implemented in Australia, as part of the Aboriginal and Torres Strait Islander Point-of-Care Testing Program since April 2020.

To eliminate any chance of aerosol or droplet generation and thus reduce some of the additional conditions above such as wearing personal protective equipment (PPE), which is in short global supply, immediate viral inactivation prior to specimen processing would be ideal. Whilst various SARS-CoV-2 inactivation protocols have been described, chemical inactivation is most suited to the POC workflow as it does not introduce new equipment or delay processes like heat and/or other inactivation methods. Several commercially available Viral Transport Mediums (VTMs) contain a chemical inactivation ingredient. These include the Primestore^®^ Molecular Transport Medium (MTM) [6], which is a United States Food and Drug Administration (FDA)-registered VTM (USA) that has been demonstrated to effectively inactivate and preserve multiple pathogens, including mycobacterium tuberculosis (MTB) [7] and influenza virus [8], and does not compromise the sensitivity of the GeneXpert platform in the detection of MTB or first-line MTB rifampicin (RIF) resistance (MTB/RIF assay) [9,10]. COPAN™, another VTM, is registered as a non-propagating transport and culture medium with the FDA but does not contain any inactivating agents [11]. To date, there have been nine studies that have evaluated chemical inactivating ingredients and shown effective virucidal activity against SARS-CoV-2 [12,13,14,15,16,17,18,19,20], but to our knowledge there has been no evaluation of commercially available VTMs that could be utilized in an appropriate workflow for community-based molecular POC testing.

The aims of our study were to evaluate the effectiveness of commercially available VTMs; two with the inactivating ingredient guanidine thiocyanate (Primestore^®^ and COPAN™ eNAT) to inactive SARS-CoV-2 virus, and a third VTM without inactivating ingredients (COPAN), alongside traditional disinfectants and heat inactivation. As guanidine thiocyanate is incompatible with bleach, a commonly used cleaning reagent in molecular testing workflows [12], we also evaluated an alternative disinfectant solution (Sterigene^®^) that is listed by the manufacturer to inactivate coronaviruses in general; there are no data to date that has specifically observed activity against SARS-CoV-2.

## 2. Materials and Methods

### 2.1. Reagents and Facilities

We utilized the sequenced SARS-CoV-2 clinical viral isolate (VIC001) [21], and permissive Vero E6 cells in culture (immortalized African monkey kidney cells, ATCC Cat# CRL-1586, American Type Culture Collection, Manassas, VA, USA). Two VTMs that contain inactivating ingredients were evaluated: Primestore^®^ MTM (Cat#PS-MTM-3, Longhorn Vaccines and Diagnostics, San Antonio, TX, USA) and COPAN™ eNAT (Cat#6C058N.RUO, COPAN Diagnostics, Murrieta, CA, USA), both containing guanidine thiocyanate (Table 1). We also included (guanidine-containing) control inactivation reagents viral lysis buffer (AVL, Cat#19073, QIAGEN LLC, Hilden, Germany). Two propagation mediums were evaluated relevant to diagnostic processing: (i) COPAN™ Universal Transport Medium (UTM; Cat#350CV, COPAN Diagnostics), which does not contain an inactivating ingredient and is the most commonly used VTM for Xpert^®^ assays, and (ii) the COPAN™ alternative HCY-CV19-UTM (Cat#CY-F003-40, Huachenyang (HCY), Shenzhen Technology Co., Ltd., Shenzhen, China). Two disinfectant methods were assessed, (i) using the chemical solution Sterigene^®^ (Oragene Global, Auckland, New Zealand) at the recommended 10% (*v*/*v*) of neat product, and (ii) without chemicals, through the use of heat inactivation. Heat is capable of effectively eliminating SARS-CoV-2 infectivity of 50 mL of cell-free SARS-CoV-2 supernatant incubated at 65 °C for 60 min [18]. These conditions were replicated to inactivate virus at the same high-titer viral concentration (0.5 × 10^7^ median tissue culture infectious dose, TCID50/mL) then plated adjacent to the positive control.

The UNSW Institutional Biosafety Committee approved SARS-CoV-2 viral expansion under physical containment level 3 (PC-/BSL-3) biosafety restrictions. All viral infections were performed within class-2 biosafety cabinets within the Kirby Institute PC3 facility under high-efficiency particulate air (HEPA) filtered 75 kpa negative pressure, and full-hooded suits connected with full face mobile support equipment (MSE) personal P3 respirators. All workers within the PC3 were nasal swab PCR SARS-CoV-2 negative prior to experiments and were periodically nasal swabbed and PCR tested to ensure no in facility viral transmission had occurred at any time when working with the virus in containment.

### 2.2. Viral Inactivation Protocols

A standardized protocol from World Health Organization (WHO) guidelines on viral inactivation using reagents was used for validation of titrated chemical compounds with potential virucidal activity [22]. In brief, 0.5 × 10^6^ (low dose) and 0.5 × 10^7^ TCID50/mL (high dose) SARS-CoV-2/VIC001 suspended in 50 μL of minimum essential medium (MEM) were immersed in 200 μL of the “test solution”, immediately diluted at 1:10 dilution using MEM to 1:10,000,000 within 2 min. Vero E6 cells were overlaid with 200 μL of the dilution series and incubated for 72 h.

Independently, BEx method was used to recover remaining viral particles present in each test reagent (Appendix A). This method was previously used to describe virucidal studies for the Ebola pandemic [23] to reduce cytotoxic effects of the reagents used for inactivation. In brief, this method retains viral particles by a filter that allows successive removal of the majority of the inactivating agent and then is replaced by a buffer that is not toxic to cells (buffer exchange) used in downstream viral outgrowth. This increases sensitivity, as higher concentrations of virus may be recovered after exposure to the inactivating agent. As above, 0.5 × 10^5–7^ TCID50/mL of SARS-CoV-2/VIC001 suspended in 50 μL of minimum essential medium (MEM) was immersed in 200 μL of the “test solution”, inverted gently five times and allowed to stand for up to 10 min. Each mixture was overlaid upon a 50 kDa Vivaspin-500 column (Cat#VS0131, Sartorius, Göttingen, Germany), then centrifuged at 12,000× *g* for 7 min. The mixture was buffer-exchanged using three washes with 500 μL of Dulbecco’s phosphate buffered saline (D-PBS, Gibco Thermofisher Scientific, Waltham, MA, USA). Viral particles remaining in the column were resuspended using 250 μL MEM with 5% fetal bovine serum (FBS). This “neat” suspension was diluted in series (1:5 fold) to a final dilution factor of 1:390,625; this five-fold dilution series was chosen to capture potential loss of product resulting from use of the buffer exchange column. Following dilution, 40 μL of each series were overlaid (in quadruplicate) upon 5000 seeded Vero E6 cells supported by 160 μL MEM per well of cell-culture grade 96-well plates (Cat#353075, Becton Dickinson, Franklin Lakes, NJ, USA). To further increase sensitivity, virions were spun onto the surface of plated cells, using a plate adapted centrifuge for 1200× *g* for 40 min at room temperature. The cells were washed three times using complete support media, consisting of minimum essential media (MEM) + 5% fetal bovine serum (FBS) and incubated in a humidified atmosphere at 37 °C and 5% CO_2_ for 72 h.

After this time, sensitivity of viable viral detection was further increased by a second viral outgrowth step, where 40 μL was transferred to a second, identical 96-well tissue-culture plate containing 20,000 suspended Vero E6 cells and incubated for a further 48 h. Viral replication for the inoculation plate at 72 h and the transfer plate (after 120 h) were detected via live cell counting in situ, using a 1/10 dilution of NucBlue Live ReadyProbes Reagent (Cat#R37605, Invitrogen, Carlsbad, CA, USA). Detection of SARS-CoV-2 RNA using the Allplex 2019-nCoV diagnostic PCR kit for nasal swabs (Cat#RP10243X, Seegene, Seoul, Korea); (E, RdRp- and N-genes) occurred via quantitative PCR (provided by NSW Health Pathology, POW-Randwick campus), as a sensitive and specific indicator of active viral replication in cell-culture samples [24].

### 2.3. Analysis and Statistics

Live cell counts were determined using an inCell-2500HTS analyzer (Cytiva/GE Healthcare, Chicago, IL, USA), by six-field imaging using orange (Brightfield) and blue (NucBlue/Hoescht 37,622 live nuclear stain, Thermofisher Scientific) filters, then stored as text-in-file-format (.tiff) uncompressed images. Viral cytopathic effect was machine-scored through reduction of nuclei counts using inCarta (Cytiva) with pre-determined nuclei and fluorescent threshold settings to enable recognition and counting of cells while accounting for background fluorescent signal. The cell count for each test well was compared to the total counts of uninfected control wells present in each plate, then normalized to 1000 cells per well using formulae generated in Microsoft Excel. Change in positive control nuclear cell count from normalized cell count was used as a measurement to test an effect of media on SARS-CoV-2 cytopathic effect. The median cell count of quadruplicate wells was calculated and compared using a *t*-test assuming Gaussian distribution (*p* < 0.05). Measurement of viral titre at 50% TCID50/mL for SARS-CoV-2 infecting Vero E6 cells was calculated using the Spearman and Kärber excel algorithm [25]. Cell counts and PCR Ct values were compiled and tabulated using Prism (GraphPad Holdings LLC, San Diego, CA, USA).

## 3. Results

### 3.1. VTMs and Reagents with Inactivating Ingredients

The WHO method showed Primestore^®^, COPAN™ eNAT and the Qiagen viral lysis buffer (AVL), which all contain guanidine thiocyanate to varying degrees, were toxic when applied directly to cells in tissue culture at high concentrations. In two separate experiments, cells were only viable at 72 h of incubation at dilutions of the reagent >1:10,000 dilution (data not shown). A reduction in toxic effect was observed when media was moved across to the transfer plate (diluted by transferring 40 μL to new cells); in this case, viable cells were observed at a 1:1000 dilution of the reagent (Figure 1C). This toxic effect was readily distinguished from SARS-CoV-2-driven cytopathic effect (CPE), based on cellular morphology using Brightfield (light) microscopy and nuclei counts (Appendix A, Column C); this effect generated curve based upon toxicity at high concentrations, which appeared equivalent to viral infectious dose values ranging between 4.46 × 10^3^ TCID50/mL for Primestore^®^ and 2.51 × 10^5^ TCID50/mL for COPAN™ eNAT (due to the inability of the inCarta software to detect nuclei; Table 1, *italics with asterisk*). There was no additional loss of cell count in the presence or absence of spiked SARS-CoV-2, which was attributed to CPE.

Figure 1A,C, show viral replication and lysis of Vero E6 cells using the WHO method at low or high viral concentrations. Nuclei counts that effectively overlap suggest there is sufficient virucidal activity for these media to inactivate SARS-CoV-2. These inactivation/toxicity data were supported by a lack of detectable SARS-CoV-2 RNA for the MTM buffers and control Qiagen viral lysis buffer. This was the case at all steps in the dilution series, and in the case of positive controls where viral infectivity was observed at up to 4.46 × 10^6^ and 1.41 × 10^8^ TCID50/mL (Table 1), which equates to a >10^7^ fold-reduction in viral infectivity.

The BEx method allows for testing virucidal activity at higher SARS-CoV-2 concentrations by eliminating the cytotoxic chemicals in Primestore^®^ and COPAN™ eNAT MTM reagents from the assay and therefore keeping cells viable. In the presence of SARS-CoV-2 at both low and high starting inoculum, no cell death was observed (Table 1, recorded as “NR” under “molecular test media”). Processing of samples using the buffer exchange method did not reduce SARS-CoV-2 infectivity. Positive control samples spiked into D-PBS at up to 0.5 × 10^7^, washed and resuspended in MEM, indicated CPE at low starting viral load of 4.7 × 10^3^ (dilutions up to 1:625, Figure 1F), and high viral load of 4.35 × 10^6^, recovering cell count at >1:78,125. This indicates both Primestore^®^ and COPAN™ eNat MTM were effective at SARS-CoV-2 inactivation at this concentration (Figure 1I). A positive (untouched) assay control was used to determine if any change in viral infectivity occurred from Vivaspin processing, by adding directly to cells. Overlapping cell counts with the buffer exchanged SARS-CoV-2 (Figure 1I,J; Table 1 BEx assay controls) indicate virus was efficiently recovered from the Vivaspin column when tested. Therefore, a minimum 3.9 × 10^5^-fold reduction in SARS-CoV-2 viral load was observed using the BEx method when incubated in Primestore^®^ MTM and COPAN™ eNAT, and the buffer-exchange (BEx) assay was capable of reporting virucidal activity.

### 3.2. VTM without Inactivating Ingredient

Using the WHO method, COPAN™ viral transport medium (VTM) was less toxic to cells (Figure 1D, dotted line). Spiking of SARS-CoV-2 into COPAN™ at 0.5 × 10^6^ TCID/mL indicated CPE at >1:10,000 dilution, and TCID50 was calculated at up to 4.46 × 10^5^ and 2.51 × 10^6^ TCID50/mL at Day 3 (data not shown) and at Day 5. Positive SARS-CoV-2 PCR in the COPAN test well up to the last dilution with observed CPE (Table 1). HCY-CV-19 UTM support media was less-tolerated by cells, requiring further dilution (Figure 1D, dotted line). When SARS-CoV-2 was spiked into HCY-CV-19 UTM, CPE was consistently observed at low viral load (Figure 1B) and high viral load, calculated at 1.41 × 10^7^ and 4.46 × 10^8^ TCID50/mL at Day 3 (data not shown) and Day 5, respectively (Figure 1D; Table 1, viral transport media). The negative control (mock infection with PBS) was the only dilution series without the presence of CPE.

### 3.3. Sterilizing Solution (Sterigene)

Sterigene^®^ was toxic to the cells in culture at a similar dilutions; the calculated toxicity for cells cultured in Sterigene^®^ spiked with SARS-CoV-2 was equivalent to viral CPE at 1.41 × 10^3^, and 7.92 × 10^2^ TCID50/mL for Day 3 and 5; these values were lower than TCID50 values for Sterigene^®^ reagent alone (data not shown), supporting evidence for complete virucidal activity against SARS-CoV-2. These results were confirmed by no PCR amplification from culture medium. It was difficult to pass Sterigene^®^ over the Vivaspin column using the BEx method, requiring an additional 10 min (i.e., 20 min at 12,000× *g*) for the last two buffer exchange centrifugation steps to entirely remove the reagent from the void volume. Nevertheless, cells were viable at all steps of the dilution curve, including when the reagent was spiked with up to 0.5 × 10^7^ TCID50/mL of SARS-CoV-2 in both the inoculation plate at up to 72 h, and the transfer plate at up to 120 h (Figure 1K).

### 3.4. Heat Inactivation

Heat-inactivation was included in these experiments as a processing control. No CPE was evident at any point in the titration (for the inoculum plate and 120h for the transfer plate), compared to the positive control, which was detected at up to 1.41 × 10^8^ TCID50/mL (Figure 1H,K). These results were validated by a lack of detectable viral amplification, indicating greater than 3.9 × 10^5^-fold reduction in SARS-CoV-2 viral load using when incubated in complete MEM support media and incubated at 65 °C for 1 h.

## 4. Discussion

These experiments demonstrate highly concentrated SARS-CoV-2 virus was eliminated following a short-term (2-min) suspension in the PrimeStore^®^ MTM and COPAN™ eNAT (including the Qiagen viral lysis buffer AVL) but not in COPAN™ or HCY-CV-19 UTM, which were only moderately cytotoxic for Vero E6 cells in culture. Secondly, we were able to demonstrate that PrimeStore^®^ MTM and COPAN™ eNAT completely inactivated SARS-CoV-2 at the highest titre that was available in the facility, which was orders of magnitude higher than clinical samples [26]. These observations were achieved through the use of a buffer exchange protocol that enabled permissive cells to remain viable despite the potential cytotoxicity of the viral inactivating agents. Similarly, heat treatment for 1 h at 65 °C eliminated viral replication, with no virus being detected in culture supernatant at the last observed well with CPE (1:1000), for the inoculation or transfer plate. We also demonstrated effective inactivation of SARS-CoV-2 using laboratory grade disinfectant (Sterigene^®^). Conversely, COPAN™ UTM and HCY-CV19 UTM in the presence of viable cells supported viral infectivity at or greater than levels observed in the positive controls.

Recent publications have provided conflicting evidence for [12], and against [18], the ability of viral lysis buffers that contain guanidine thiocyanate to deactivate SARS-CoV-2, specifically including the Qiagen AVL buffer that we used in our experiments. It is clear from our results that there was inactivation of SARS-CoV-2 by Qiagen AVL at up to a minimum 5 log_10_, at 2 min incubation, and up to 7 log_10_ at 10 min incubation at the highest available viral titre to infect permissive cells. Differences in virucidal activity for this buffer may result from methodological differences in inactivation and/or detection methods. Pastorino et al. [18] adopted a standardized analytical virucidal testing methodology (Analytice, reference document number: ANFOR NF EN 14476 + A2), which resulted in detection of a positive SARS-CoV-2 signal, following addition of bovine-serum albumin (BSA at 3 g/L), ethanol or Triton X-100; these inactivation steps were also prior to culture of supernatant for up to 10 days post-inactivation.

One consideration when using media that contain guanidine thiocyanate in molecular workflows is their well-documented incompatibility with hypochlorite solutions (i.e., bleach). Solutions containing >70% ethanol (v/V) are recommended for cleaning and decontamination of spills when handling SARS-CoV-2 [15]. However, situations may arise where the supply or use of these disinfectant solutions may be difficult (e.g., cleaning large surface areas) and other reagents are chemically incompatible (i.e., release of toxic fumes from cleaning chemicals that contain ammonia or sulfates). Here, we demonstrate that Sterigene^®^ (a mixture of halogenated tertiary amine and organic salts, polymeric biguanide hydrochloride surface active agent’s corrosion inhibitor, chelating agents, stabilising agents and demineralized water) diluted to the working concentration (10% *v*/*v*) is a suitable alternative for cleaning SARS-CoV-2 processing facilities.

A strength of our study is using the BEx methodology, and its reproducibility with automated collection of CPE data through machine counting of nuclear staining. The advantage of the BEx method over the standard WHO methodology is largely due to the capacity to remove reagents that may interfere with cell substrate viability, resulting in reduced ability of any remaining active virus to infect viable cells. This has advantages over the WHO method, as it has greater sensitivity in subsequent viral outgrowth and detection. Yet, the short coming to BEx is the time taken to remove the inactivating solution from the viral culture and minimum inactivation times greater than 10 min. In contrast, the WHO method enables observation of shorter inactivation time, as the rapid dilution of the virus from the inactivation solution can enable determination of minimal inactivation times within 2 min. Using both the WHO and BEx methodologies, we could determine not only the minimal inactivation time but also could increase the stringency of the viral detection.

Limitations to the methodologies applied here are as follows: (i) the chemical or reagent used needs to be compatible with the Vivaspin polyethersulfone membrane, due to additional spin time being required for Sterigene^®^ buffer exchange, and (ii) reagents applied directly using the WHO method are themselves toxic to cells. While it is crucial to have a reagent-only control, a higher starting viral inoculum may also be required to observe a readout of viral pathogenic effect, over that of reagent cytotoxicity. Establishing a dilution series at a 10-fold, rather than 5-fold, dilution may also be more appropriate in this instance, or if the transfer plate were to be incubated for more than 72 h.

We had originally intended for this publication to evaluate the effectiveness of three VTM to inactivate SARS-CoV-2 (Primestore^®^, COPAN™ eNAT and HCY-CV19-UTM). It was our understanding from the advertising material and an invoice from the company that the HCY-CV19-UTM (Cat# CY-F003-40) provided as the part of the iclean VTM kit (by Shenzhen Technology Co., Ltd.) was capable of inactivating SARS-CoV-2. Our results demonstrate that HCY-CV19-UTM shows absolutely no sign of viral inactivation, unlike our observations of the other VTMs tested, and should not be used for the purpose of increasing operator safety. From our results, HCY-CV19-VTM could in fact decrease operator safety if there is a false belief it does inactive the virus and appropriate barriers are not implemented.

Our findings provide strong evidence that PrimeStore^®^ MTM and COPAN™ eNAT eliminated SARS-CoV-2 infectivity at very high concentration and within 2 min while conserving the integrity of the viral nucleic acid for RT-PCR testing [27]. Integration of these MTMs into POC testing protocols will eliminate any aerosol generation of SARS-CoV-2 and could reduce the need for additional conditions set by WHO, such as additional PPE, which are in limited supply. An important consideration when using media that contain guanidine compounds is incompatibility with hypochlorite solutions (bleach). Therefore, it is recommended that 70% alcohol solutions or Sterigene be used as a replacement for decontamination of spills when handling SARS-CoV-2.

## Figures and Tables

**Figure 1 viruses-12-01208-f001:**
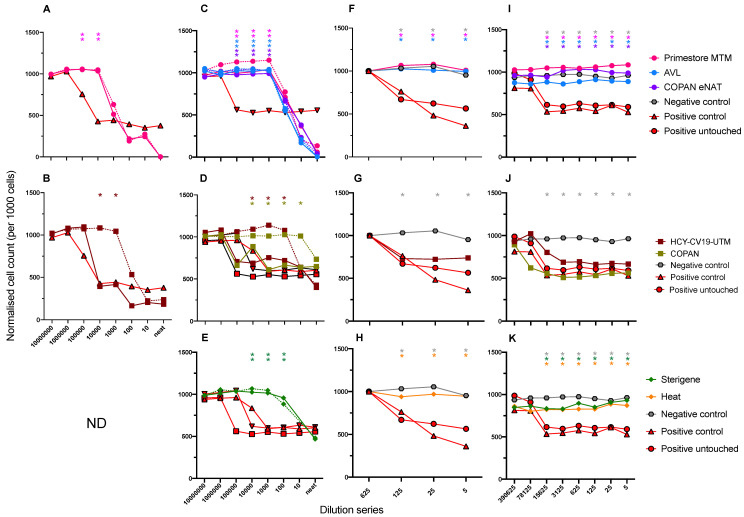
Normalized viable cell count (per 1000) at Day 5 of culture following neutralization of SARS-CoV-2 in viral transport media at serial dilution. Count of viable Vero E6 cells is using NucBlue stain (normalized to a count of 1000 cells), when using the WHO method at low viral load (**A**,**B**), and high viral load (**C**–**E**), diluted in 10-fold series. Viable cell count at Day 5 culture following neutralization and recovery of SARS-CoV-2 using the BEx method is indicated at low viral load (**F**–**H**), and high viral load (**I**–**K**), diluted in five-fold series. Incubation in media is indicated by colour (grey hexagon: negative control; red square/triangle(up)/triangle (down): positive [control(triplicate)], circle (pink): Primestore^®^ MTM, (purple) COPAN™ eNAT, (blue) Qiagen AVL, square (khaki) COPAN™ UTM, (maroon) HCY-CV-19UTM, diamond (green) Sterigene^®^, diamond (gold) heat-inactivation). Incubation in media only for the WHO method is indicated by the dotted line connecting data points. Significant differences in median cell count (of quadruplicate measurements) above cell counts observed for control wells positive for SARS-CoV-2 are represented by asterisks corresponding to the colour of each medium at each point of serial dilution (faded asterisks indicate incubation in media alone; ND, test not done).

**Table 1 viruses-12-01208-t001:** Effect of media upon SARS-CoV-2 infection as measured by viral load count (TCID50/mL), averaged nuclear cell count, cytopathic effect and detection of viral products by PCR (values are sampled from WHO (1:1000) and the buffer-exchange (BEx) method assay (1:625). Cytopathic effect (CPE) and PCR results (reported as binary +/−) at 1:1000 dilution of the positive control culture and SARS-CoV-2 viral detection in culture as above PCR assay threshold for target genes [E-, RdRp- and N-genes], sampled at last sample where CPE was observed in dilution of the test reagent. (ND, not done; red text, highlighting where a positive SARS-CoV-2 infection was observed; NR, no positive result recorded; italic text with asterisk (*), toxic effect was observed from the reagent, resulting in a false positive TCID50 reading).

Assay	Test	Condition	Viral Load (TCID50/mL)	Averaged Viable Cells Counted	Cytopathic Effect (CPE)	SARS-CoV-2 (PCR Detectable)
A (WHO)	Assay controls	Positive control	1.41 × 10^8^	591	+	+
		Positive (assay) control—untouched	ND	551	+	+
	Toxicity controls	Primestore MTM	*4.46 × 10^3^* *	1150	−	−
		COPAN (eNAT)	*7.92 × 10^3^* *	1019	−	−
		Kit control (AVL)	*7.92 × 10^3^* *	1025	−	−
		COPAN (UTM)	*1.41 × 10^2^* *	1010	−	−
		CV19-UTM	*7.92 × 10^2^* *	1140	−	−
	Molecular test media	Primestore MTM	4.46 × 10^3^	1046	−	−
		Kit control (AVL)	7.92 × 10^3^	1040	−	−
		COPAN (eNAT)	7.92 × 10^3^	991	−	−
	Viral transport media	COPAN (UTM)	2.51 × 10^6^	610	+	+
		CV-19 (UTM)	4.46 × 10^8^	753	+	+
B (BEx)	Assay controls	Negative control	NR	974	−	−
		Positive control	4.35 × 10^6^	575	+	+
		Positive (assay) control—untouched	2.90 × 10^6^	630	+	+
	Molecular test media	Primestore MTM	NR	1046	−	−
		Kit control (AVL)	NR	912	−	−
		COPAN (eNAT)	NR	1033	−	−
	Viral transport media	COPAN (UTM)	4.36 × 10^6^	514	+	+
		CV-19 (UTM)	1.94 × 10^6^	692	+	+

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
