# Peer review of "Evaluation of Commercially Available Viral Transport Medium (VTM) for SARS-CoV-2 Inactivation and Use in Point-of-Care (POC) Testing"

_viruses, 2020, doi:10.3390/v12111208_

Round 1
Reviewer 1 Report
This manuscript is clearly timely, and consequently it is key the scientific community work together to ensure good, rigorous science is applied to COVID
I am not an expert in virology, so I am being entirely honest that as a highly trained scientist, I have genuine suggestions on a couple of bits. these comments are the ones that I am passing onto the editor too. I am battling COVIDtesting issues myself, plus my own reseach +academia. I do want to help.
1) VTM of any kind can impact upon viral detection. its a thorough job in this manuscript, but samples need to be confirmed. ie, the testing is essential of enriched virus (purchased) rather than saliva or nasopharyngeal swab. its key that this is done, becuase POC optimisation needs to be done using POC setting. I entirely believe the work done, but need to see a patient sample doesnt impact it too. you could mix saliva with your viral (COV negative etc) to show that VTM+a bodily fluid isnt going to impact this too.
2) its confusing as hell to explore the datatables vs graphs. coudld significance and error please be presented on the traces?
Author Response
1) VTM of any kind can impact upon viral detection. its a thorough job in this manuscript, but samples need to be confirmed. ie, the testing is essential of enriched virus (purchased) rather than saliva or nasopharyngeal swab. its key that this is done, becuase POC optimisation needs to be done using POC setting. I entirely believe the work done, but need to see a patient sample doesnt impact it too. you could mix saliva with your viral (COV negative etc) to show that VTM+a bodily fluid isnt going to impact this too.
As part of the Australian government response to the COVID pandemic, Primestore MTM is now being used in the network, with positive and negative COVID-19 cases successfully identified and all positive cases confirmed with alternative laboratory platforms. These results indicate successful POC testing on a broader level than demonstrated here. Positive and negative QC testing is being conducted monthly, with results as expected; inclusive of the QC samples made from patient samples, dried onto NPS swabs and placed in MTM before testing. Further research is underway to assess stability of different VTMs stored at various temperatures and time periods to ensure reproducible testing from all sites within the POC testing network. These results will be released for public knowledge at a future time.
2) its confusing as hell to explore the datatables vs graphs. coudld significance and error please be presented on the traces?
We thank the reviewer for their concern regarding the reporting of results. In response to this, the (trace) graphs (Figure 1, 2a, 2b) have been condensed to a single figure (Figure 1), and the two tables (Tables 1 and 2) condensed to a single Table (Table 1), both including representative experimental data from Day 5. Any other values of note for the results are entered as text. We have included a test measuring significant change from a normal median cell count (1000 cells), resulting from virucidal activity observed in the positive controls. Values for every data point in the trace graphs are representative of quadruplicate wells, so reporting the error range is possible. However, it was strongly felt that the addition of these error bars (with multiple curves already present on the same graph) would detract from clear interpretation of data.
Reviewer 2 Report
Evaluation of commercially available viral transport medium (VTM) for use in SARS-CoV-2 point-of-care (POC) testing.
The Authors present a comprehensive and detailed manuscript investigating the virucidal inactivation potential of various commercially available viral transport mediums. The aim is to advise readers which options provide a safer handling potential for in-field POC testing.
The title of the manuscript does not convey the intent of the study and should reflect that primary aim is to assess inactivation of virucidal activity/titre reduction. This needs to be incorporated into the title.
The manuscript is very long and clearly contains a significant amount of work, however its length and number of figures detract from the findings and the message intended to be delivered. This presentation could be significantly reduced by condensing tables and figures and summarizing in a much more succinct manner. The abstract achieves this, but the manuscript is a labored read and the figures/tables all appear to essentially show the same data in different formats.
The content of this manuscript is of interest, utility and should be presented concisely to achieve its intended outcome of informing users that various commercially available virus transport mediums will provide a biosafety barrier when handled subsequently.
A comparison of each method result in a table would be a clear and obvious way to allow the reader to quickly and easily compare the performance of the products. It is not easy in its current form.
Suggestions
Condense considerably to a shorter communication article.
Table to be significantly simplified & condensed
Figure 1 – day 3 not necessary
Figure 2a & b – buffer exchange - can be omitted and provided as text
Figure 3 – cpe and detail is not necessary.
A good reference as a clear and concise way of presenting similar data is
Author Response
The Authors present a comprehensive and detailed manuscript investigating the virucidal inactivation potential of various commercially available viral transport mediums. The aim is to advise readers which options provide a safer handling potential for in-field POC testing.
The title of the manuscript does not convey the intent of the study and should reflect that primary aim is to assess inactivation of virucidal activity/titre reduction. This needs to be incorporated into the title.
Thank you for the advice, the title has been revised to the following:
Evaluation of commercially available viral transport medium (VTM) for SARS-CoV-2 inactivation and use in point-of-care (POC) testing
The manuscript is very long and clearly contains a significant amount of work, however its length and number of figures detract from the findings and the message intended to be delivered. This presentation could be significantly reduced by condensing tables and figures and summarizing in a much more succinct manner. The abstract achieves this, but the manuscript is a labored read and the figures/tables all appear to essentially show the same data in different formats.
The content of this manuscript is of interest, utility and should be presented concisely to achieve its intended outcome of informing users that various commercially available virus transport mediums will provide a biosafety barrier when handled subsequently.
A comparison of each method result in a table would be a clear and obvious way to allow the reader to quickly and easily compare the performance of the products. It is not easy in its current form.
Suggestions
Condense considerably to a shorter communication article.
Table to be significantly simplified & condensed
Figure 1 – day 3 not necessary
Removed
Figure 2a & b – buffer exchange - can be omitted and provided as text
Figure 3 – cpe and detail is not necessary.
Removed to supplementary material
Thank you for the suggestions, in response and for the sake of brevity to convey a clear scientific message, we have removed Table 1 and condensed the data into a singular representation of detectable viral load for each method described. This data will now be representative under the column name (Viral load) and a central focus in Table 2, alongside the existing cell-count and PCR detection columns which will hereafter be re-labelled Table 1.
Figure 1 will be amended to contain Day 5 data only, which is sufficient to convey the message of the paper, being that MTM is effective at inactivation of SARS-CoV-2. We feel that it is important to convey the effect of each methodology on capacity to measure the viral load curve, and especially to convey the cytotoxic effect of media. For this reason, the BEx Day 5 data we have decided to include adjacent to the WHO Day 5 data.
Figures 2a and 2b will therefore be written as text; a critical portion of that data has been merged into the new Figure 1. Figure 3 (microscopic images), while we agree with the reviewer is repeating existing data in another form, it is important to convey the clearly different effect of the two methodology upon the readout for each assay. In response, we feel it is best to relocate the Figure 3 to a supplementary material Figure S1.
Round 2
Reviewer 1 Report
I’m happy with the changes and response given
Reviewer 2 Report
The Authors have significantly shortened the manuscript in line with suggestions made on first review. This is now located in the supplemental section. They have also changed the title to reflect the intent of the manuscript.
A simplified table/figure for readers to very quickly identify what buffers were effective would still benefit this paper, without having to look through and decipher all the tables and graphs to determine this.
The Authors have been careless in editing and have not re-read the manuscript. Starting Line 216 is a sentence disconnected from its original sentence, seemingly from line 210. It is unclear what is intended here.
At line 238, 239, 240 the values 0.5x10^7, 4.7x10^3, 4.35x10^6 are titers of virus and should be expressed as such (TCID50/ml) – as they are in other locations in the manuscript.
Line 317 remove ‘for’ (Sterigene as for the molecular transport media…)